# Membrane Fouling and Electrochemical Regeneration at a PbO_2_-Reactive Electrochemical Membrane: Study on Experiment and Mechanism

**DOI:** 10.3390/membranes12080814

**Published:** 2022-08-22

**Authors:** Liankai Gu, Yonghao Zhang, Weiqing Han, Kajia Wei

**Affiliations:** 1Key Laboratory of Jiangsu Province for Chemical Pollution Control and Resources Reuse, School of Environment and Biological Engineering, Nanjing University of Science and Technology, Nanjing 210094, China; 2School of Environmental Science and Engineering, Yancheng Institute of Technology, Yancheng 224051, China

**Keywords:** membrane fouling, reactive electrochemical membrane, PbO_2_ catalytic layer, antifouling, regeneration

## Abstract

Membrane fouling and regeneration are the key issues for the application of membrane separation (MS) technology. Reactive electrochemical membranes (REMs) exhibited high, stable permeate flux and the function of chemical-free electrochemical regeneration. This study fabricated a micro-filtration REM characterized by a PbO_2_ layer (PbO_2_-REM) to investigate the electro-triggered anti-fouling and regeneration progress within REMs. The PbO_2_-REM exhibited a three-dimensional porous structure with a few branch-like micro-pores. The PbO_2_-REM could alleviate Humic acid (HA) and Bisphenol A (BPA) fouling through electrochemical degradation combined with bubble migration, which achieved the best anti-fouling performance at current density of 4 mA cm^−2^ with 99.2% BPA removal. Regeneration in the electro-backwash (e-BW) mode was found as eight times that in the forward wash and full flux recovery was achieved at a current density of 3 mA cm^−2^. EIS and simulation study also confirmed complete regeneration by e-BW, which was ascribed to the air–water wash formed by bubble migration and flow. Repeated regeneration tests showed that PbO_2_-REM was stable for more than five cycles, indicating its high durability for practical uses. Mechanism analysis assisted by finite element simulation illustrated that the high catalytic PbO_2_ layer plays an important role in antifouling and regeneration.

## 1. Introduction

Membrane separation (MS) is one of the most efficient and applicable techniques in wastewater treatment due to its advantages of high efficiency, small footprint, easy automation, and requiring relatively little maintenance [1,2,3]. MS can selectively remove pollutants by changing its pores, e.g., particle filtration (PF), microfiltration (MF), ultrafiltration (UF), nanofiltration (NF), reverse osmosis (RO), and etc. However, although the application of MS is increasingly growing, there still exist many challenges, the most critical of which is membrane fouling. Fouling is inevitable because the rejected pollutants are left in the membranes and finally block the pores along with the treatment. The traditional way to solve this critical problem is pretreatment (including chemical oxidation or pH adjustment) or regeneration (physical or chemical cleaning) [4]. Apparently, both of them are independent units that operate separately from MS, which prolong the treatment time and increase cost. Notably, even though the regeneration can be achieved by physical or chemical cleaning, the service life of the membrane will be shortened as these operations might damage its structure. It is believed that improving the antifouling ability or providing a self-cleaning function to MS comprise a “green way” to solve this problem. Moreover, pollutants often simultaneously exist in the water bodies, making it very difficult and costly to apply traditional MS. Therefore, it is urgent to functionalize MS, which could reject and meanwhile eliminate pollutant.

Recently, introducing electricity (applying an electric field or directly supplying current) into membrane techniques has attracted lots of attention because it can efficiently address fouling challenges and is meanwhile conducive to membrane regeneration [5,6,7,8,9]. Electrical functions, including electrochemical oxidation (EO)/reduction, electrostatic, electrophoresis, and electroporation, can degrade, transform, and/or migrate pollutants to alleviate or even eliminate membrane fouling. Especially, coupling MS and EO as an integrated technology has been intensely studied in recent decades, which can concurrently separate and degrade pollutant, and is named as reactive electrochemical membrane (REM) [10,11,12,13]. REMs exhibited strong antifouling ability and stable permeate flux when operated and therefore are now successfully employed and in water pollution control including treatment, disinfection, and purification, and show good performance [14,15,16]. On the other hand, EO processes are often accompanied by the side reaction of oxygen evolution. Although oxygen evolution reactions might interfere with the degradation of pollutants, they are helpful in migrating pollutant. More importantly, they are beneficial for regeneration because, together with the flow, the generated oxygen could form air–water washing, which is well-known as much better than solely water washing.

The catalytic layer plays the most important role on the electrochemical performance of REMs, which as reported to be RuO_2_, PbO_2_, SnO_2_, and Ti_4_O_7_, etc. [17,18,19,20]. Among these, the PbO_2_ layer was most prominent because it possesses higher oxygen evolution potential (OEP) than RuO_2_, has a longer service life than SnO_2_, and is more stable than Ti_4_O_7_ in anodic conditions. Some researchers questioned the leach of Pb^4+^ when using PbO_2_ in environmental applications, but this could be avoided as the PbO_2_ layer was fabricated with TiO_2_ nanotube arrays (NTAs) [21,22]. Moreover, the manufacture of the PbO_2_ layer was mature and cheap. Therefore, PbO_2_ layers possess a better balance between high catalytic efficiency, durability, and scalability, making them suitable layers for REM in wastewater treatment. However, previous work mainly studied PbO_2_-REM in the view of electrochemical electrodes and focused on the degradability of pollutants, few studies evaluated it as a membrane (e.g., antifouling ability and regeneration). Recently, Jing and his co-workers studied membrane fouling and electrochemical regeneration of a sub-stoichiometric TiO_2_ REM, and the results suggested that this REM can provide an efficient and cost-effective regeneration scheme [23,24]. Nevertheless, the fabrication of the sub-stoichiometric TiO_2_ REM was highly conditioned and costly (high temperature of 1050 °C and at reduced atmosphere of H_2_). Thus, it is necessary to study the antifouling and regeneration of PbO_2_-REM that could replace sub-stoichiometric TiO_2_ REM in water treatment applications in the future.

In this study, a PbO_2_-REM was fabricated by anodic oxidation and the electrodeposition method. Characterization of the REM was undertaken by FE-SEM, EDS, and a pore-size analyzer. Antifouling ability of the REM was evaluated by target pollutants of Humic acid (HA) and Bisphenol A (BPA). Regeneration was studied by the comparation of electro-/none forward wash mode and electro-/none backwash mode. Repeated regeneration tests in electro-backwash (e-BW) mode were also conducted to measure the durability of the REM. Especially, the mechanism of antifouling and regeneration of PbO_2_-REM was discussed with the assistance of electrochemical impedance spectroscopy (EIS) and a simulation (by FLUENT model) study.

## 2. Materials and Methods

### 2.1. Reactant and Material

All chemicals used in the experiments were reagent grade or higher without further purification. Lead nitrate (Pb (NO_3_)_2_), Nitric acid (HNO_3_), Sodium fluoride (NaF), Ammonium fluoride (NH_4_F), ethylene glycol ((CH_2_OH)_2_), and sodium sulphate (Na_2_SO_4_) were purchased from Shanghai Aladdin Biochemical Technology Co., Ltd. Sulphuric acid (H_2_SO_4_), Sodium hydroxide (NaOH), Isopropanol (C_3_H_8_O), Potassium ferricyanide (K_3_Fe(CN)_6_), Potassium ferrocyanide trihydrate (K_4_Fe(CN)_6_ · 3H_2_O), Humic acid (HA), and Bisphenol A (BPA) were purchased from Sinopharm group. Porous titanium (Ti) substrate with a diameter of 30 mm and thickness of 2 mm was obtained from Bairong Co., Baoji, China; the purity was greater than 99%. Ultrapure water was obtained from Milli-Q^®^ EQ 7000.

### 2.2. Fabrication of PbO_2_-REM

Fabrication of PbO_2_-REM was carried out according to a combination of our previous works [25,26,27,28]. Briefly, porous Ti substrate was cleaned by sodium hydroxide, isopropanol, and pure water in order to remove the impurities before the fabrication. The TiO_2_ NTAs was prepared by anodic oxidation in ethylene glycol containing 9.15 vol% water (weight ration = 2) and 0.3 wt% NH_4_F while stainless steel was used as a counter electrode at 60 V for 2 h. After this, the PbO_2_ layer was formed by electrodeposition in the electrolyze containing 0.5 M Pb (NO_3_)_2_, 0.1M HNO_3_, and 0.04 M NaF with the current density of 0.03 A cm^−2^ at 60 °C for 60 min. The PbO_2_-REM was fabricated by the abovementioned processes and finally cleaned by ultrapure water and dried at 100 °C for 5 min.

### 2.3. Characterization of PbO_2_-REM

The FESEM images were taken by a field emission scanning electron microscope (JEM-2100F, JEOL, Japan) with primary electron energy of 20 KV. The EDS data were obtained by an energy dispersive spectrometer (X-MaxN 80T IE250, Oxford, UK). The pore size and porosity were analyzed by a bubble point analyzer (PSDA-30MU, GaoQ, Nanjing, China). EIS measurement was performed by an electrochemistry workstation (CHI-700, Chenhua, China) in a three-electrode system using PbO_2_-REM as the working electrode, platinum electrode as the counter electrode, and Ag/AgCl electrode as the reference electrode. All the tests were undertaken in the electrolyte consisting of 5 mM K_4_Fe(CN)_6_ and 5 mM K_3_Fe(CN)_6_ with an amplitude of 5 mV in the sinusoid perturbation and over a frequency range of 100 kHz to 10 mHz.

### 2.4. Experimental Apparatu and Methods

The whole system for the experiments is shown in Figure 1. Briefly, the system was composed of a REM reactor, DC power, and a diaphragm pump. The reactor was a plexiglass installation assembled by PbO_2_-REM, stainless steel, O-ring, and electric wire (copper). Herein, PbO_2_-REM worked as the membrane/anode and stainless steel worked as the cathode, which was separated by an O-ring with the distance of 10 mm and connected to DC power by electric wire. A diaphragm pump was used to control flow under a trans-membrane pressure of 0.1 Mpa. HA was used as the target pollutant because its structure is similar to the natural organic matter (NOM) found in drinking water sources. BPA was used to evaluate the oxidation performance of PbO_2_-REM. The concentrations of HA and BPA were 150 and 20 mg L^−1^, respectively, and they must be freshly prepared before the experiments. The concentration of BPA was measured by an ultraviolet spectrometer (UV-2450, Shimadzu, Japan) at 278 nm. Na_2_SO_4_ was chosen as the electrolyte to ensure the conductivity with a concentration of 5 g L^−1^. During the experiments, the fluid flowed through the stainless steel cathode and PbO_2_-REM anode in sequence, using a DC power supply to apply current. All experiments were operated in this mode expect the backwash, which was oppositely operated. Both forward wash and backwash as well as its stages are abbreviated by pure water to PW, forward wash to FW, backwash to BW, electro-forward wash to e-FW, electro-backwash to e-BW, fouling stage to FS, and post treatment to PT in the subsequent discussions.

### 2.5. FLUENT Model

The simulation study for regeneration by the e-BW mode was conducted by FLUENT software. To simplify the calculation, the model was designed into a rectangular area with a width of 22 μm and a length of 23 μm, the pore size was controlled at 6 μm (which was close to the average pore size of the PbO_2_-REM), and the distance between the pore was 2 μm while the length of the pore was 15 μm. The flushing water was inlet from the bottom and outlet from the top. The simulation parameters for PW-BW were set as follows: inflow velocity of 10^−5^ m s^−1^, initial pollutant concentration of 100 mol m^−3^, and diffusion coefficient of 10^−14^ m^2^ s. The simulation parameters for air–water washing were set as follows: oxygen production of 500 mg m^−2^ s^−1^, oxygen density of 1.32 kg m^−3^, oxygen inlet velocity of 3.922 × 10^−4^ m s^−1^, and bubble radius of 10^−6^ m; other parameters are the same as PW-BW.

## 3. Results and Discussion

### 3.1. Structure and Morphology of PbO_2_-REM

#### 3.1.1. Surface Morphology and Microstructure

FESEM images of the original porous Ti substrate and PbO_2_-REM are shown in Figure 2. It can be seen in Figure 2a that the porous Ti substrate exhibited a three-dimensional structure with numerous micro-pores; these pores formed the tunnel of the membrane. It can be seen that TiO_2_ NTAs were fabricated onto the surface of substrate, which was compact and uniform; a higher magnification image of Figure 2c demonstrates that the pore size of TiO_2_ NTAs was between 60~80 nm. This stable structure of TiO_2_ NTAs creates the perfect conditions for PbO_2_ electrodeposition. After electrodeposition, as we can see in Figure 2d,e, the PbO_2_ layer was generated, which was completely covered the porous Ti substrate. Figure 2f shows us the high-magnification image of PbO_2_ layer, and it can be observed that the layer exhibited typical PbO_2_ crystal structure without a nano-scale porosity, which is in accordance with related studies [21,29]. The results in Figure 2 indicated that TiO_2_ NTAs and the PbO_2_ layer were successfully formed on porous Ti substrate, which means the preparation of PbO_2_-REM is complete and can be used for the later experiments.

#### 3.1.2. Element and Pore Size Distribution

The EDS of PbO_2_-REM is shown in Figure 3a. It can be found in Figure 3a that the elements of PbO_2_-REM are Pb and O. Especially, it also can be found that the Ti element was not detected, indicating that the deposited PbO_2_ layer possesses a large thickness that completely covered the Ti substrate, which is in accordance with the result found in the SEM study. In order to obtain insight into porosity, the pore size distribution of the PbO_2_-REM was tested as shown in Figure 3b. It can be seen that the pore size of PbO_2_-REM ranged from 4 to 18 μm. The inset shows us that most of pore the sizes were between 6 and 11 μm. However, it is unexpected that the result of the bubble point analyzer was not highly in accordance with the results of SEM analysis, which were smaller. This can be explained by the fact that the bubble point analyzer can only test the equivalent diameter of the narrowest part of pores [17]. Therefore, the visible and large pore at the surface of PbO_2_-REM in SEM study was not monitored in Figure 3b. Despite this, it can be verified in the pore-size distribution study that PbO_2_-REM maintained the three-dimensional porous structure of porous Ti substrate that can be used as a membrane for microfiltration.

### 3.2. Antifouling Ability of PbO_2_-REM

The result of antifouling experiments under different current densities of 0, 1, 2, 3, and 4 mA cm^−2^ can be found in Figure 4. As is shown in Figure 4a, the flux dramatically dropped to 16.1% after 10 min of experiment without current and then continued to decrease with the final flux of 0.13 %, indicating the REM was seriously fouled. After applying current, the decline of flux was alleviated and more flux was maintained. Other studies also obtained the same outcome, this mainly being attributed to the self-cleaning function given by EO that the foulant can be degraded by ·OH or migrate by bubbles [30,31,32]. The inset shows the total wastewater permeated under different conditions; it is clear that more wastewater was permeated after applying current and the maximum volume belongs to 3 mA cm^−2^. Interestingly, it also can be found that the performance of 2, 3, and 4 mA cm^−2^ had nearly no difference. This indicates that increasing current had no significant influence when it was higher than 2 mA cm^−2^. This is attributed to the HA used in this study mostly consisting of an insoluble substance, which was non-degradable by EO. Although high current density could generate more oxygen bubbles to remove the accumulated pollutant, abundant bubbles will occupy the space for the fluid and therefore hinder permeation. Furthermore, high current density also leads to part of the electricity transferring to thermal energy, causing lost energy [33]. Nevertheless, the results shown in Figure 4b illustrated that high current density is good for organic pollutant removal; the BPA concentration was extremely low (0.17 mg L^−1^) after 90 min treatment under the condition of 4 mA cm^−2^. Therefore, considering the toxicity of treated wastewater, the current density should be chosen as 4 mA cm^−2^. Consequently, although PbO_2_-REM cannot eliminate the rejected insoluble pollutants, it can effectively remove soluble pollutants (e.g., BPA) due to its excellent electrochemical properties.

### 3.3. Regeneration of PbO_2_-REM

#### 3.3.1. Regeneration in FW Mode

The results of the regeneration experiment in FW mode are shown in Figure 5a. The experiment consisted of FS (0–90 min), PW-FW (90–140 min), e-BW (140–240 min), and PT (240–300 min). It can be obtained that the normalized flux rapidly decreased to 18.24% during the first 10 min of FS, then decreased to 5.7% after 20 min and only 0.91% left at the end of FS (90 min), suggesting that the REM had been completely blocked. After FS, the PbO_2_-REM was simply cleaned by the pure water for 5 min and reassembled in the reactor for e-FW. As we can see, the flux recovered to 7.22% at the beginning but still gradually decreased to 4.03% at the end of PW-FW. Furthermore, in order to recover more flux, e-FW was operated under oxidation conditions with the current density of 1 mA cm^−2^. During the e-FW, the flux quickly recovered to 16.02% but still experienced a decrease within 150 to 180 min; the final recovery of the flux was 10.34%. The last 60 min of PT experiment shows us that flux can only be recovered by 11.27% after PW-FW and e-FW, indicating that this mode cannot regenerate the HA fouled PbO_2_-REM. The reason is that there existed an electrode potential drop with depth into the pores as reported by reference [23]. In this case, the interior space of PbO_2_-REM has much lower potential compared with the surface that is not available to produce ·OH. Therefore, only direct oxidation functions are not as effective as ·OH. In addition, the shear force of flow was too weak to help regeneration in this mode because of the significant increase of the trans-membrane pressure, as the REM was completely blocked due to the HA fouling.

#### 3.3.2. Regeneration in BW Mode

Since the FW mode cannot regenerate HA fouled PbO_2_-REM, regeneration in the BW mode was evaluated. The experiment consisted of FS, PW-BW, and e-BW as is shown in Figure 5b. First of all, it can be observed that regeneration in BW mode was obviously superior to that in the FW mode (eight times under the same current density). The inset shows the change of flux during the fouling stage; the final flux of this process was 0.75%. Furthermore, it can be observed that the flux recovered to 42.5% after PW-BW. Although this is much better than that in FW, it is still not enough for regeneration. Due to this, e-BW was further undertaken independently under five different current densities of 1, 2, 3, 4, and 5 mA cm^−2^ for 60 min. As shown, the flux was significantly recovered by e-BW for 82.9% and 95.6%, corresponding to 1 and 2 mA cm^−2^. Li and co-workers also found an almost complete restoration of Ti_4_O_7_ REM by e-BW [34]. Obviously, the recovery in this stage was significant and the higher current density applied, the better regeneration conducted. By further increasing the current density to 3, 4, and 5 mA cm^−2^, it can be found that the flux was fully recovered to 106.2%, 100.1%, and 101.3%, indicating the HA fouled REM was completely regenerated. Notably, it can be observed that the limitation of recovered flux was obtained under the condition of 3 mA cm^−2^ because there is no more increase on flux when current density is higher than this condition.

Moreover, an EIS study was also conducted to verify the full recovery of the fouled membrane by the electro-BW mode, and the data were provided as Nyquist plots (seeing in Figure 5c). It can be seen that the resistance Ohms of fouled REM was much higher than that of the original and after e-BW, this being mainly ascribed to the adsorption of HA that decreases the rate of charge transfer [24]. In addition, it can also be found that the resistance Ohms of the original and after e-BW are very close, indicating that the rate of charge transfer was regenerated, and so as to the membrane.

Herein, it was assumed that the excellent performance of PbO_2_-REM on regeneration at the e-BW mode might be attributed to the synergistic effect of flushing flow and generated oxygen. In this case, air–water washing was formed, which could chase away the foulant easily. To verify this conjecture, a simulation study was carried out in the BW mode with or without air as is shown in Figure 6. The colors shown in the figure represent the concentration of foulant; thus, the denser red zone it is, the more serious the fouling is. Furthermore, it can be found from the upper figures that the red zone was faded by PW-BW with the time, but it was inefficient because there were still some red zone left even at 1000 ms, indicating that PW-BW cannot fully recover the fouled membrane, which is in accordance with the result in Figure 5b. On the contrary, it was clearly shown in the lower figures that the red zone was almost completely faded at 500 ms when oxygen (air) was involved (namely air–water washing). More significantly, it also can be observed that the red zone on this condition was fading faster than that of pure water cleaning at any time. These results implied that EO plays an important role in regeneration, and the mechanism is deeply discussed in Section 3.4.

#### 3.3.3. Repeated Regeneration in e-BW Mode

In order to evaluate the durability of PbO_2_-REM, repeated regeneration in e-BW mode was undertaken as is shown in Figure 7. The experiments were operated by 90 min FS and 60 min e-BW under the condition of 3 mA cm^−2^ for five cycles; the results are summarized in Table 1. As is shown, the flux was reduced to extremely low for all five cycles after FS of 0.77%, 0.81%, 0.68%, 0.17%, and 0.41%, respectively. Furthermore, the flux can be significantly recovered by the e-BW mode. Especially, all the fluxes were up to 100% after regeneration and the best performance belonged to the 2^nd^ cycle, which was 111.30%. These results indicated that PbO_2_-REM was stable in the e-BW mode and could adapt to practical and long-term employment.

### 3.4. Mechanism of Antifouling and Regeneration of PbO_2_-REM

Based on the previous description, the PbO_2_-REM performed well on antifouling and regeneration, which is mainly attributed to the EO function from the high catalytic efficiency of the PbO_2_ layer. In order to obtain a deeper understanding, herein, we conducted a discussion of the mechanism of antifouling and regeneration of PbO_2_-REM. First of all, it should be pointed out that both the degradation by ·OH and bubble migration by oxygen evolution contribute to the removal of foulant. For antifouling, the foulant adsorption and deposition were hindered due to the large amount of ·OH by the PbO_2_ layer and the enhanced shear force by the generated oxygen bubble together with the transmembrane microflows, which is in accordance with a previous study [35]. For regeneration, besides the electrochemical behaviors mentioned above, the potential distribution of REM also affects its performance. Therefore, Jing et al. suggested that e-BW was more efficient due to the following reasons: (1) foulants will move from low-potential regions to high-potential regions where more ·OH exists, (2) dissociation of the foulant is easier to achieve through the shear force by BW flow, and (3) shear force can be enhanced by generated gas bubbles and the subsequent bubble collapse [24]. In this study, a more specific demonstration of the regeneration process of REM by the e-BW mode was made to be complementary to the former study. As we can see in Figure 8a, the REM was completely blocked after fouling; even though water-flow backwash (Figure 8b) could remove some of the foulant, there were still many foulants left in the blind side of the pore. Figure 8c shows us the electrochemical behaviors of the PbO_2_ layer after applying the current. It can be observed that the foulant was first decomposed to the debris and thus chased away to the middle of the pore by the oxygen bubbles because the bubbles were generated at the wall of the pore. In this case, air–water wash was electrochemically triggered by the generated bubbles and flushing flow that can easily remove the foulant as shown in Figure 8c. Finally, the foulant was completely removed at the end of e-BW, which means the REM was regenerated. As a conclusion, the mechanism discussed above illustrated that the excellent performance of PbO_2_-REM on antifouling and regeneration, mainly attributed to the synergistic effect of electrochemical behaviors of PbO_2_ layer that can simultaneously degrade and migrate the foulant.

## 4. Conclusions

In this paper, PbO_2_-REM was successfully fabricated through anodic oxidation and electrodeposition. Characterization of PbO_2_-REM suggested that the REM exhibited a porous structure with micro-pores, indicating its ability to achieve microfiltration when working as a membrane. We also highlighted the high performance of the PbO_2_-REM on antifouling and regeneration through a series of experiments. The decline of flux due to the fouling of HA could be alleviated and the BPA could be efficiently removed by applying current. The fouled REM can be completely regenerated in the BW mode whereas FW cannot. The EIS study also proved that the e-BW mode could completely regenerate the fouled membrane, while the simulation study indicated that oxygen evolution plays an important role. Moreover, the repeated regeneration test indicated that the fabricated REM has high durability. The mechanisms of antifouling and regeneration were discussed as well to obtain a better understanding of the electrochemical behaviors of PbO_2_-REM during its operation. The main reason for the excellent performance of PbO_2_-REM was its superior electrochemical properties, which could degrade pollutant by generating a large amount of ·OH and the enhanced shear force to the migrated pollutant by forming bubbles during the process. Therefore, this study indicated that the fabricated PbO_2_-REM might have good prospects in wastewater treatment.

## Figures and Tables

**Figure 1 membranes-12-00814-f001:**
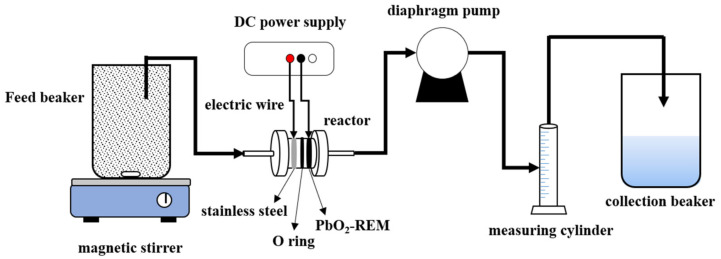
Schematic diagram of PbO_2_-REM and system for the experiments.

**Figure 2 membranes-12-00814-f002:**
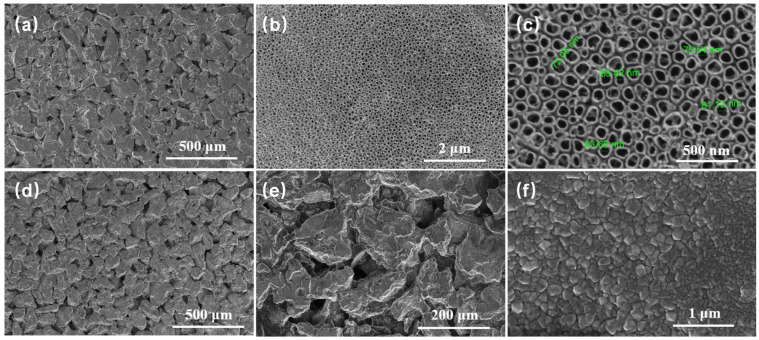
FESEM images of porous Ti substrate and PbO_2_-REM: (**a**) porous Ti substrate (×400), (**b**) TiO_2_ NTAs (×50,000), (**c**) TiO_2_ NTAs (×200,000), (**d**) PbO_2_ layer (×400), (**e**) PbO_2_ layer (×600), and (**f**) PbO_2_ layer (×100,000).

**Figure 3 membranes-12-00814-f003:**
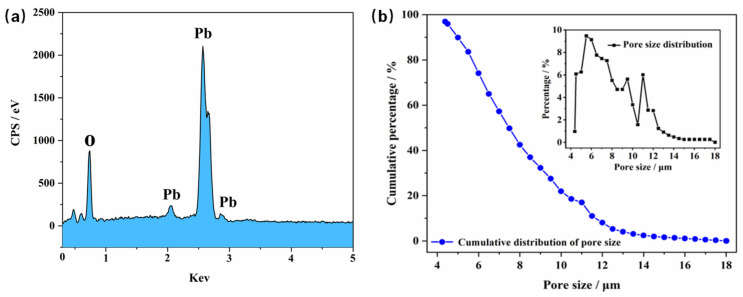
(**a**) EDS spectrum of PbO_2_-REM; (**b**) pore size distribution of PbO_2_-REM.

**Figure 4 membranes-12-00814-f004:**
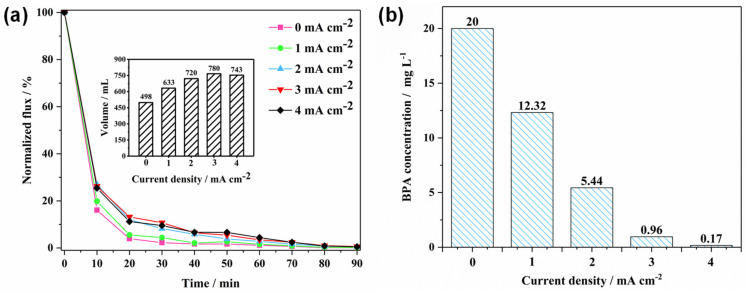
(**a**) Normalized flux of PbO_2_-REM of the antifouling experiment under different current densities and (**b**) BPA concentration after treatment under different current densities.

**Figure 5 membranes-12-00814-f005:**
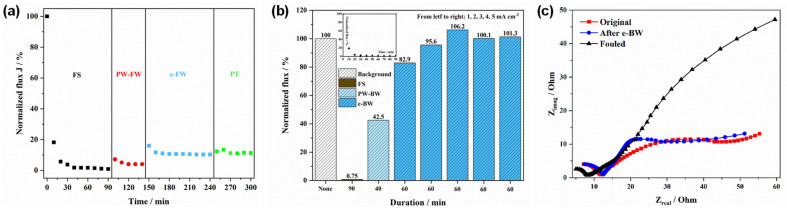
Normalized flux of PbO_2_-REM of the regeneration experiment in (**a**) FW mode, (**b**) BW mode, and (**c**) EIS spectra of original REM, the REM after electro-BW, and fouled REM in the complete frequency range.

**Figure 6 membranes-12-00814-f006:**
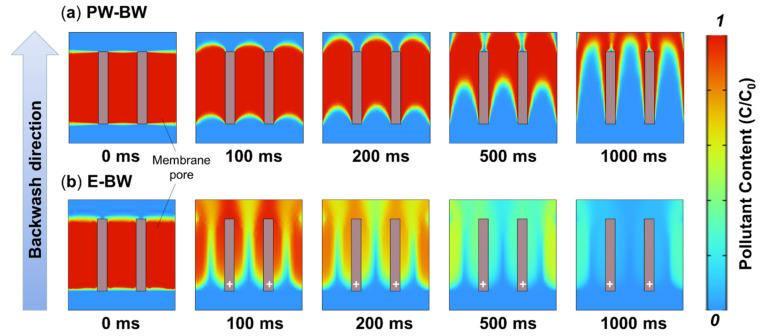
Simulation results obtained by FLUENT. Conditions and parameters can be founded in Section 2.5.

**Figure 7 membranes-12-00814-f007:**
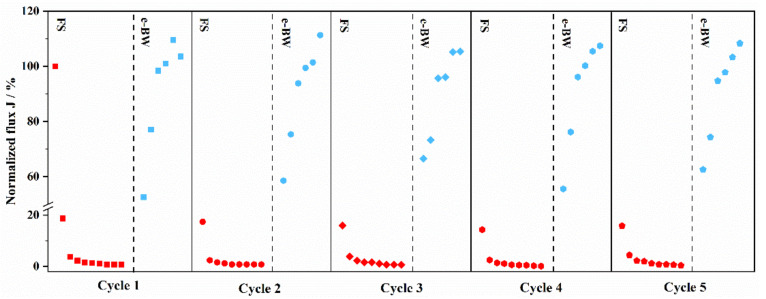
Normalized flux of PbO_2_-REM of the regeneration experiment in the repeated e-BW test.

**Figure 8 membranes-12-00814-f008:**
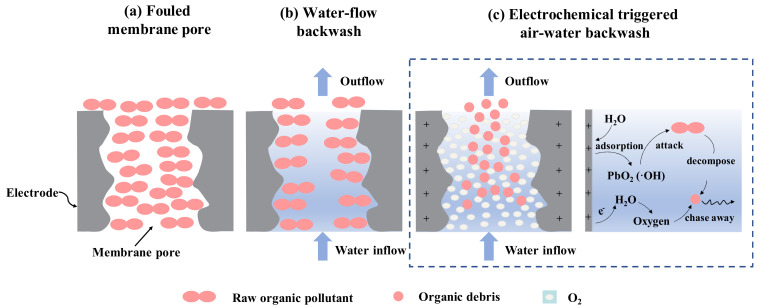
Schematic diagram of the mechanism of (**a**) fouled membrane pore, (**b**) PW-BW, (**c**) electrochemical triggered air–water wash with the electrochemical reaction process.

**Table 1 membranes-12-00814-t001:** Normalized flux of PbO_2_-REM for repeated regeneration in e-BW mode.

Cycle	Normalized Flux (%)	
	Fouling	Regeneration
1st	0.77	103.57
2nd	0.81	111.29
3rd	0.68	105.43
4th	0.17	107.48
5th	0.41	108.31

## Data Availability

Not applicable.

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
