# Peer review of "Membrane Fouling and Electrochemical Regeneration at a PbO2-Reactive Electrochemical Membrane: Study on Experiment and Mechanism"

_membranes, 2022, doi:10.3390/membranes12080814_

Round 1

Reviewer 1 Report

There is no doubt that this is a detailed study, which fabricated a mi-cro-filtration REM characterized with PbO2 layer (PbO2-REM) to investigate the electro-triggered anti-fouling and regeneration progress within REMs. Focus on membrane fouling, reactive electrochemical membrane, PbO2 catalytic layer, antifouling and regeneration. Overall, this work is very meaningful, but there are still some details, and please refer to the following questions and suggestions for improvement. I suggested that it should be accepted after a minor revise.

1. Page 2, line 83-93. The objectives of this study should be expressed in more concise and organized terms.

2. Page 3, line 130. It is suggested that Section 2.4. Schematic diagram of the experiments apparatu should be changed to Experimental apparatu and methods or other headings that correspond to the content of this section.

3. Page 4, line 137-138. “HA was used as target pollutant because its structure is similar with natural organic matter (NOM) found in drinking water sources”. Was the concentration change of HA recorded and how was it measured?

4. Page 4, line 143-144. “During the experiments, the fluid was flow through stainless steel cathode and PbO2-REM anode in sequent and using a DC power supply to apply current”. What is the value of the applied current or voltage?

5. In Section 3, it is suggested to add some discussion content with other studies.

6.Page 7, line 237-240. “Therefore, only direct oxidation functioned which was not as effective as ·OH. In addition, the shear force of flow was too weak to help regeneration in this mode because the significant increase of trans-membrane pressure as the REM was completely blocked due to the HA fouling.” How did you come to this conclusion?

7. Moderate English (language and style) improvement is required, for example,

a. Page 4, line 172. “Figure 2f shows us…”, Other forms are suggested to replace first person pronouns.

b. Page 10, line 338. “In this paper, we successfully fabricated PbO2-REM through anodic oxidation and electrodeposition.” Rearrange the grammar of this sentence, preferably without the first person.

Author Response

There is no doubt that this is a detailed study, which fabricated a micro-filtration REM characterized with PbO2 layer (PbO2-REM) to investigate the electro-triggered anti-fouling and regeneration progress within REMs. Focus on membrane fouling, reactive electrochemical membrane, PbO2 catalytic layer, antifouling and regeneration. Overall, this work is very meaningful, but there are still some details, and please refer to the following questions and suggestions for improvement. I suggested that it should be accepted after a minor revise.

Answer: The authors appreciate the reviewer’s positive comments and have revised our paper in the best way as we could. Please see specific replies below.

1. Page 2, line 83-93. The objectives of this study should be expressed in more concise and organized terms.

Answer: Thank you for the suggestion. The objectives have been revised in concise and organized terms (Page 2, line 83-91).

“In this study, a PbO2-REM was fabricated by anodic oxidation and electrodeposition method. Characterization of the REM was undertaken by FE-SEM, EDS and pore size analyzer. Antifouling ability of the REM were evaluated by target pollutant of Humic acid (HA) and Bisphenol A (BPA). Regeneration was studied by the comparation of electro-/none forward wash mode and electro-/none backwash mode. Repeated regeneration test in electro-backwash (e-BW) mode was also conduct to measure durability of the REM. Especially, mechanism of antifouling and regeneration of PbO2-REM was discussed with assistant of electrochemical impedance spectroscopy (EIS) and simulation (by FLUENT model) study.”

2. Page 3, line 130. It is suggested that Section 2.4. Schematic diagram of the experiments apparatu should be changed to Experimental apparatu and methods or other headings that correspond to the content of this section.

Answer: Thank you for the suggestion. Schematic diagram of the experiments apparatu have been revised into Experimental apparatu and methods (Page 4, line 138).

3. Page 4, line 137-138. “HA was used as target pollutant because its structure is similar with natural organic matter (NOM) found in drinking water sources”. Was the concentration change of HA recorded and how was it measured?

Answer: Thank you for the suggestion. The major difference between two target pollutant is their molecular radius which HA could accumulate in the pores (and eventually blocked pore) whereas BPA cannot. Considering this, it is more directly to judge fouling, anti-fouling and regeneration by flux rather than concentration of HA. Therefore, the concentration change of HA were not recorded. Moreover, the reason why concentration of BPA was measured is mainly because it reflects the degradation ability of PbO2 layer.

4. Page 4, line 143-144. “During the experiments, the fluid was flow through stainless steel cathode and PbO2-REM anode in sequent and using a DC power supply to apply current”. What is the value of the applied current or voltage?

Answer: Thank you for the suggestion. The value of applied current was given in form of current density as was discussed in section 3.2, 3.3 and illustrated in Figure 4a and 5b. (Page 5, line 204; Page 6, line 224; Page 7, line 257; Page 7, line 266).

5. In Section 3, it is suggested to add some discussion content with other studies.

Answer: Thank you for the suggestion. We have added some discussion content with other studies in section 3.2 and 3.3. (Page 6, line 207-209; Page 7, line 261-262).

“Other studied also obtained same outcome, this mainly attributed to self-cleaning function given by EO that the foulant can be degraded by ·OH or migrate by bubbles [30-32].” and “Li and co-workers also founded almost complete restoration of Ti4O7 REM by e-BW [34]”

6. Page 7, line 237-240. “Therefore, only direct oxidation functioned which was not as effective as ·OH. In addition, the shear force of flow was too weak to help regeneration in this mode because the significant increase of trans-membrane pressure as the REM was completely blocked due to the HA fouling.” How did you come to this conclusion?

Answer: Thank you for the suggestion. the conclusion can be supported by reference [24] at page 514, 515 and 516. For example, “Results indicate that the potential drops drastically with depth into the porous membrane and the depth at which the potential is able to drive ·OH production is only approximately 0.023 μm.”, “However, due to potential drop with distance with in the membrane only direct electron transfer reactions can occur”, “However, after regeneration the EIS data suggests further fouling of the support layer, likely due to the polymerization of organic fragments removed from the active layer and flushed into the support”.

7. Moderate English (language and style) improvement is required, for example,

Page 4, line 172. “Figure 2f shows us…”, Other forms are suggested to replace first person pronouns.

Page 10, line 338. “In this paper, we successfully fabricated PbO2-REM through anodic oxidation and electrodeposition.” Rearrange the grammar of this sentence, preferably without the first person.

Answer: Thank you for the suggestion. We have checked the manuscript and replaced all first-person pronouns, meanwhile, some grammar errors were also revised. (Page 3, line 116; Page 3, line 133; Page 4, line 173; Page 4, line 175-176; Page 5, line 187; Page 5, line 204-205; Page 6, line 232-233; Page 7, line 280; Page 10, line 364).

Author Response

I think it is of great interest in the community of membrane science, because Membrane fouling and regeneration are the key issues to the application of Membrane separation (MS) technology. As a result, I will recommend the publication of this manuscript in the current form.

Answer: The authors appreciate the reviewer’s positive comments, it is a great encouragement to us.